# Unsupervised Monocular Depth and Camera Pose Estimation with Multiple Masks and Geometric Consistency Constraints

**DOI:** 10.3390/s23115329

**Published:** 2023-06-04

**Authors:** Xudong Zhang, Baigan Zhao, Jiannan Yao, Guoqing Wu

**Affiliations:** 1School of Information Science and Technology, Nantong University, Nantong 226019, China; zhang.xd69@ntu.edu.cn (X.Z.); wgq@ntu.edu.cn (G.W.); 2School of Mechanical Engineering, Nantong University, Nantong 226019, China; yaojiannan2016@ntu.edu.cn

**Keywords:** depth estimation, camera pose, visual odometry, unsupervised learning

## Abstract

This paper presents a novel unsupervised learning framework for estimating scene depth and camera pose from video sequences, fundamental to many high-level tasks such as 3D reconstruction, visual navigation, and augmented reality. Although existing unsupervised methods have achieved promising results, their performance suffers in challenging scenes such as those with dynamic objects and occluded regions. As a result, multiple mask technologies and geometric consistency constraints are adopted in this research to mitigate their negative impacts. Firstly, multiple mask technologies are used to identify numerous outliers in the scene, which are excluded from the loss computation. In addition, the identified outliers are employed as a supervised signal to train a mask estimation network. The estimated mask is then utilized to preprocess the input to the pose estimation network, mitigating the potential adverse effects of challenging scenes on pose estimation. Furthermore, we propose geometric consistency constraints to reduce the sensitivity of illumination changes, which act as additional supervised signals to train the network. Experimental results on the KITTI dataset demonstrate that our proposed strategies can effectively enhance the model’s performance, outperforming other unsupervised methods.

## 1. Introduction

Estimating scene depth and camera pose from video sequences is a critical topic in visual perception and forms the foundation of many advanced tasks. Such estimations can be used to build 3D scene structures, which can be implemented in various industrial environments, including autonomous driving, visual navigation, and augmented reality [1,2,3]. Traditional methods rely on geometric cues in the image for inference, making them sensitive to challenging environments with low texture or strong lighting changes [4,5,6,7,8]. Conversely, learning-based depth and pose estimation methods exhibit better adaptability to challenging environments [9,10,11,12,13]. These methods take the image sequence as input and output the depth map and camera pose through a nonlinear mapping formed by the collaboration of neurons. This is similar to the human brain processing high-dimensional information after observing with human eyes. Supervised learning-based methods primarily use exact sensors like differential GPS, LiDAR, and IMU to obtain labeled data and then train the estimation network on these labeled data to learn the mapping function by minimizing the difference between the model’s predicted values and the label values.

While supervised methods have shown excellent performance, their reliance on massive quantities of labeled data during model training poses a significant limitation. Acquiring labeled data in real-world scenarios often requires expensive equipment or large amounts of manpower, thus preventing them from further improving the performance of their models [9]. In contrast, unsupervised methods do not require expensive labels during training and can leverage larger datasets and more complex models to achieve better performance [10]. This is exemplified by recent models such as ChatGPT, which uses large amounts of data and parameters to achieve state-of-the-art performance [14]. Researchers have also proposed numerous unsupervised methods for tasks related to depth recovery and pose estimation [10,11,12,13]. A common principle is using the view synthesis process to generate supervised signals to train the model. This technique employs two sub-networks based on convolutional neural networks (CNNs), which respectively estimate the depth map of a view (target view) and the camera pose of the view and the adjacent view (source view). Once the depth and camera pose is estimated, the source view can be projected onto the target view to synthesize a new view, and the entire framework is then trained by minimizing the photometric error between the synthesized new view and the target view.

View synthesis necessitates a static scene without occluded areas; however, real-world scenes are rife with dynamic objects and occlusions, inevitably resulting in unstable training [15]. Consequently, numerous studies have proposed various masks to mitigate outliers in the scene during the view synthesis process [15,16,17,18,19,20,21]. However, these methods often overlook the impact of dynamic objects and occlusion areas on pose estimation, jointly trained with depth estimation during the view synthesis process. As a result, any degradation in pose estimation accuracy will decrease depth estimation accuracy. Furthermore, training the entire unsupervised framework primarily relies on the photometric differences between the synthesized view and the target view, implying that when the illumination changes drastically or when the training sequence is lengthy, inconsistent illumination intensities may interfere with the model’s learning process.

To overcome these challenges, we propose a new unsupervised learning framework for estimating scene depth and camera pose. Due to the fact that both dynamic objects and occluded regions in the scene can simultaneously affect the computation of the loss function and the estimation of the camera pose, we have designed multiple masks to identify different types of outliers in the scene during forward propagation. These computed masks are combined and applied in two ways: firstly, to prevent outliers from participating in calculating the loss function during view synthesis, and secondly, as a supervised signal for MaskNet, a neural network trained to estimate outliers such as dynamic objects and occluded regions. The mask obtained by MaskNet is then used to preprocess the input of the pose network. This involves multiplying the regions that may be outliers by a small weight coefficient, thereby avoiding the influence of outliers in the scene on pose estimation. Moreover, since training the network solely with view synthesis is prone to be affected by changes in lighting, we propose several geometrically consistent loss functions, such as flow consistency, depth consistency, and pose consistency loss, to exploit the geometric properties of the 3D scene and further strengthen our model’s functionality as a whole.

The main contributions of the work are twofold: (1) We introduce multiple mask techniques to mitigate the adverse impact of outliers in the scene during the view synthesis process. Additionally, we employ a MaskNet network to address the detrimental effects of outliers on pose estimation; (2) we propose several geometric consistency constraints to alleviate the limitations of sole training with photometric consistency. Finally, we evaluate our model on the widely used KITTI dataset and demonstrate its superiority over other unsupervised methods.

The remainder of the paper is structured as follows: Section 2 provides an overview of the existing literature concerning depth and pose estimations. Section 3 elaborates on the method and the enhanced strategies proposed in this paper. The effectiveness of our approach is validated through experimental results and ablation studies, which are presented in Section 4. Finally, Section 5 offers a summary of our work.

## 2. Related Works

Visual Simultaneous Localization and Mapping (VSLAM) [4,5,6] and Structure from Motion (SFM) [7,8] are two examples of traditional geometry-based techniques used to estimate scene depth and camera pose. Both of these methods require extracting hand-designed key points from the image and matching them to estimate the camera pose, followed by the triangulation technique to estimate the scene’s structure. However, the traditional methods may fail in low-texture regions where key points cannot be extracted, presenting a challenge. Learning-based methods have emerged as a solution to overcome this challenge. These methods are classified as supervised or unsupervised based on whether or not they use labels for training.

### 2.1. Supervised Learning of Scene Depth and Camera Pose

Supervised depth recovery and camera motion estimation approaches typically consider these separate tasks. They learn them individually by minimizing the discrepancies between the estimated values and the related ground truth. The pioneering work by Eigen et al. [22] demonstrated the use of deep CNNs to predict depth from a single image using two network stacks: one for global prediction and the other for local refinement. In contrast, Liu et al. [23] employed hierarchical conditional random fields and a super-pixel pooling method to improve the quality of the depth map. Regarding the problem of camera pose estimation, also called visual odometry (VO), optical flow is a widely used technique in learning-based VO methods [24,25,26], where the optical flow (OF) field contains geometric motion. Costante et al. [25] used a self-encoder to learn the optical flow field’s low-dimensional latent feature space. Then they used this feature space to regress the camera’s 6-dimensional pose, thus improving the robustness of the estimated model. Zhao et al. [26] also used optical flow as the input to the pose estimation network and continued the work of [25] by adding recurrent neural networks for sequence learning to improve the estimation accuracy further. Despite the promising performance of supervised methods, their utility is limited by the requirement for labeled datasets, which can be arduous, costly, and prone to a lack of generalizability.

### 2.2. Unsupervised Learning of Scene Depth and Camera Pose

Garg et al. [11] pioneered a novel method to reduce the dependence on labeled data by training a network with stereo image pairs as input. The objective function of the training is to minimize the photometric discrepancies between the left image and its corresponding right synthetic image using epipolar geometric inference for view synthesis. Godard et al. [12] extended this methodology by incorporating the left-right consistency constraint for depth estimation. Additionally, Zhan et al. [13] continued to follow this approach of using stereo images as input, solving the absolute scale problem of VO estimation [6]. Meanwhile, they also proposed a feature reconstruction loss to strengthen the training of the framework.

Nevertheless, camera calibration in stereo systems is complex, leading Zhou et al. [15] to propose an approach relying on monocular sequences. Their core idea is to train both the depth estimation network and the pose estimation network by using the photometric consistency loss generated by the view synthesis process as the objective function. Building upon this pioneering work, subsequent studies have made significant strides. For example, in [27], Mahjourian et al. further considered the 3D geometry of the whole scene and required that the estimated 3D point cloud be consistent across the continuous images. Similarly, [28] utilized the 3D-2D correspondence constraint and deployed it on an autonomous driving platform via 5G telephony and wireless communication.

Despite its potential, the monocular unsupervised method has two significant limitations. Firstly, it cannot provide global scale consistent pose estimation, and secondly, the photometric consistency loss assumes a static scene without occlusion regions and dynamic objects. For scale ambiguity, Bian et al. [18] introduced geometric consistency loss to cope with scale inconsistencies on different samples, resulting in VO results comparable to the stereo image training model. Sun et al. [29] proposed two constraints that operate on predicted depth and relative poses, enforcing consistency across different training samples and jointly promoting pose and depth estimation. For dynamic objects and occlusion areas in the scene, an intuitive method is to design a mask to remove these outliers to avoid their adverse effects on the calculation of reconstruction loss. Therefore, researchers have proposed a variety of mask-generation methods. Some of them generate masks by adding networks, such as the explainability mask [15] proposed by Zhou et al., the uncertain map [16] proposed by Klodt et al., and the confidence mask [17] proposed by Chen et al. Alternatively, some methods generate masks through computation, such as those in [18,19,20,21]. In [18], Bian et al. calculated depth inconsistency to generate a self-discovered mask. Zhao et al. [19] generated an outlier elimination mask by analyzing the consistency of forward and backward optical flows. Wang et al. [20] derive overlap and blank masks during forward calculation, while Jiang et al. [21] generate a mask based on the assumption that the outlier reconstruction error is significantly greater than the average photometric error. These methods have a positive impact on reconstruction loss computation.

Additionally, several studies have explored the potential benefits of jointly training the subtasks of scene depth, camera pose, and optical flow by exploiting their inherent geometric correlation, thus allowing for better nonlinear solutions when constraints are added to the loss function [30]. For instance, Yin et al. [31] proposed a collaborative learning framework to estimate all three subtasks and reconstruct a view containing both static and dynamic scenes using the geometric relationships between them. Zhang et al. [32] introduced an optical flow estimation network and added additional supervised signals to the training of the framework through multi-view synthesis to improve the overall estimation accuracy. Based on this joint estimation framework, Zou et al. [33] also proposed optical flow consistency loss for rigid regions in the scene to improve their results. Finally, Ranjan et al. [34] added a motion segmentation task to the above three tasks, and the individual performance of each task was enhanced by joint training.

Although these methods have shown considerable advancements in the basic models, their treatment of changing scenes is limited to calculating the loss function in view synthesis, ignoring the impact on VO estimation. Moreover, the detrimental effects on training when the photometry is inconsistent are not considered.

## 3. Methods

This paper aims to learn the depth and camera pose from unlabeled monocular video sequences while exhibiting good robustness to outliers in the scene, such as dynamic objects and occlusion areas. Our framework utilizes training samples composed of three consecutive frames, with the middle frame as the target view and the other two as the source view, to obtain two target-source image pairs. Since the operations performed on the two image pairs are the same, we only show the process for one image pair, as shown in Figure 1. Our method comprises four sub-networks: (1) DepthNet, which estimates the depth map of a single image; (2) FlowNet, which estimates the optical flow between adjacent frames, is an off-the-shelf model that does not require training; (3) MaskNet, which estimates a mask containing outliers based on two adjacent frames, and its supervised signal mainly comes from the calculation of multiple masks in the view synthesis process; and (4) PoseNet, which uses the mask-preprocessed optical flow as input and outputs the camera motion.

We train the DepthNet, MaskNet and PoseNet sub-networks jointly with a final loss function comprising four parts: a photometric consistency loss, which is the primary supervised signal used during network training; a depth smoothness loss, which ensures the smoothness of the estimated depth values; a mask loss, which enhances the model’s performance by training a mask to preprocess PoseNet’s input; and a geometric consistency loss, which provides an additional weak supervised signal by adding several constraints to the model. Therefore, the final loss function can be expressed as
(1)$$L=\sum_{l}(L_{ph}^{l}+\lambda_{s}L_{sm}^{l}+\lambda_{m}L_{m}^{l}+L_{g}^{l}),$$
with $L_{ph}^{l}$, $L_{sm}^{l}$, $L_{m}^{l}$ and $L_{g}^{l}$ denoting the photometric consistency loss, smoothness loss, mask loss, and geometric consistency loss, respectively, $\lambda_{s}$ and $\lambda_{m}$ representing the corresponding weight values. The parameter *l* is the scale factor of different image sizes, and similar to previous work [15], the DepthNet outputs four depth estimation maps at different scales, with the loss function being calculated for each scale separately.

### 3.1. Photometric Consistency Loss and Smoothness Loss

The primary supervised signal for training the entire framework is derived from the photometric consistency loss generated during view synthesis. The process is illustrated in Figure 2. Given two consecutive input frames (*I_t_*, *I_t_*_+1_), represented as the target view and the source view, The DepthNet estimates the depth map of the target view. Using this depth information (*D_t_*), a pixel in the image can be projected onto a 3D point cloud as follows:(2)$$Q_{t}^{i,j}=D_{t}^{i,j}K^{-1}{\left[i,j,1\right]}^{T},$$
where *K* is the intrinsic camera matrix and $Q_{t}^{i,j}$ is the 3D point. The camera motion *T_t_* estimated by the PoseNet is used to transform $Q_{t}^{i,j}$ to the coordinates at frame *t* + 1 via $Q_{t+1}^{i,j}=T_{t}Q_{t}^{i,j}$. This 3D point can then be transformed to the camera coordinates at frame *t* + 1 using the intrinsic camera matrix *K*. Therefore, by using these projection transformation relations, we can get the projection relationship between the coordinates of the target view and the source view, which is expressed as:(3)$${\left[\hat{i},\hat{j},1\right]}^{T}=KT_{t}\left(D_{t}^{i,j}K^{-1}{\left[i,j,1\right]}^{T}\right).$$

To reconstruct the target view *I_t_*, we need to obtain the corresponding pixel values of the reconstructed frame ${\hat{I}}_{t}$ based on the projected position on frame *I_t_*_+1_. However, the pixel values after projection are not integers. Hence we use bilinear interpolation to obtain the pixel value at *p_t+_*_1_. Specifically, we linearly interpolate the 4-pixel values (top-left, top-right, bottom-left, and bottom-right) around *p_t+_*_1_ using the formula: (4)$${\hat{I}}_{t}\left(p_{t}\right)=\sum_{i\in \left\{t,b\right\},j\in \left\{l,r\right\}}w^{ij}I_{t+1}\left(p_{t+1}^{ij}\right),$$
where $w^{ij}$ is the proportional term for bilinear interpolation, measuring the spatial proximity of *p_t_*_+1_ and $p_{t+1}^{ij}$ with $\sum_{i,j}w^{ij}=1$. Then, the synthesized view ${\hat{I}}_{t}$ is obtained.

Assuming that the photometric values of 3D spatial points projected onto the target view and the source view are equal, it can be theoretically deduced that the photometric values of the target view and the synthesized target view obtained through interpolation using the source view should be consistent. This property is used to construct the loss function for training the entire framework. Similar to other works [20,21], we use the combination of the L1 norm and the structural similarity index measure (SSIM) [35] to construct the photometric consistency loss, which is expressed as: (5)$$L_{ph}\left(I_{t},{\hat{I}}_{t}\right)=\alpha \frac{1-SSIM\left(I_{t},{\hat{I}}_{t}\right)}{2}+\left(1-\alpha \right){\left\|I_{t}-{\hat{I}}_{t}\right\|}_{1},$$
where α is set to 0.85 empirically.

During the training of the depth estimation model, the edge smoothness loss is often used to filter out incorrect predictions and preserve clear details. We use the same loss function as [31], expressed as
(6)$$L_{sm}=\sum_{p_{t}}\left|\nabla D\left(p_{t}\right)\right|{\left(e^{-\left|\nabla I\left(p_{t}\right)\right|}\right)}^{T}$$
where $\left|\cdot \right|$ denotes the elementwise absolute value, ∇ represents the vector differential operator, and *T* denotes the transpose of image gradient weighting.

### 3.2. Calculated Mask and Mask Loss

Due to the many assumptions involved in view synthesis, such as the scene is static and devoid of dynamic objects and occlusions, training images that violate these assumptions will inevitably hinder model training. Therefore, it is necessary to consider the influence of these factors when using the view synthesis process to calculate the loss function. A common principle is to design masks to shield these outlier regions and prevent them from participating in the calculation of the loss function, thereby improving the performance of the model.

During the view synthesis process, when the pixels in the target view are projected onto the source view, some pixels will be projected beyond the imaging plane of the source view so that the pixel value of the point cannot be reconstructed. As shown in Figure 3, suppose there are two points ($p_{t}^{1}$, $p_{t}^{2}$) in the target view *I_t_*. When the camera moves to the frame *I_t_*_+1_, the projection of $p_{t}^{1}$ onto source view *I_t_*_+1_ is $p_{t+1}^{1}$, and the projection of $p_{t}^{2}$ is $p_{t+1}^{2}$, which is located outside the boundary of *I_t_*_+1_, making it impossible to determine the pixel value of that point accurately. Therefore, we mark all these points projected outside the boundary as 0, thus generating a boundary mask denoted as *M_e_*. After being multiplied by the target view, the mask can avoid calculating the loss function for some boundary points.

In addition to the boundary points that will affect the training of the model, the occluded areas in the scene will also affect the training of the model. For example, as shown in Figure 4a, when a car equipped with a camera travels from frame *I_t_* to frame *I_t_*_+1_, there is an obstacle on the left side of the yellow car. In frame *I_t_*, the camera can capture the areas of *S*_1_ and *S*_2_, but in frame *I_t_*_+1_, the area of *S*_1_ will be occluded by the area of *S*_2_, and only the area of *S*_2_ can be seen. Thus, the problem arises that the areas of *S*_1_ and *S*_2_ captured by frame *I_t_* is projected onto frame *I_t_*_+1_ and overlap in coordinates, making it impossible to use the image captured at frame *I_t_*_+1_ to restore the area of *S*_1_ captured at frame *I_t_*. 

When occluded areas appear in the image, they must be marked to avoid their participation in calculating the loss function. As shown in Figure 4b, when two pixels ($p_{t}^{1}$, $p_{t}^{2}$) in frame *I_t_* are projected onto frame *I_t_*_+1_, if they fall in the same grid, that is, they have four identical interpolation points, we consider occlusion to have occurred. According to the distance between these two pixels and the camera, we mark the farther point as 0 and the closer point as 1. In Figure 4b, we assume that $p_{t}^{1}$ is closer, so it is marked as 1, while $p_{t}^{2}$ is marked as 0. This generates a mask denoted as *M_o_*, which is used to identify the occlusion area.

Due to the pixel points that meet the hypothesis, the photometric errors will gradually converge to a lower value during training. In contrast, for some pixel points caused by various adverse factors, the photometric error value will always remain at a higher value [21]. Using this characteristic, those points with photometric errors much higher than the average are identified as outliers. Precisely, for each pixel on the target view, we can determine whether it is an outlier point according to the following expression: (7)$$M_{a}\left(I_{t},{\hat{I}}_{t}\right)=\left\{\begin{array}{c} 1,L_{ph}\left(I_{t},{\hat{I}}_{t}\right)\leq \beta {\bar{L}}_{ph}\left(I_{t},{\hat{I}}_{t}\right)\\ 0,L_{ph}\left(I_{t},{\hat{I}}_{t}\right)>\beta {\bar{L}}_{ph}\left(I_{t},{\hat{I}}_{t}\right) \end{array}\right.,$$
where ${\bar{L}}_{ph}$ is the average value of all photometric errors, $M_{a}$ indicates mask, β is the corresponding weight that reflects the tolerance for outliers. The larger the value, the more outliers will be retained. 1.5 is used here empirically.

The three masks mentioned above can effectively remove the majority of outliers. However, there are still some special cases that need to be considered, such as objects moving at speed similar to the camera or scenes where the camera is stationary, both of which violate the assumptions required for the view synthesis process, i.e., a moving camera and a stationary scene. To address this issue, we adopt the auto-making technique used in [12], i.e., the photometric error of the synthesized target view should be less than the photometric error calculated directly using the source view. The mask denoted as *M_s_* is expressed as follows:(8)$$M_{s}\left(I_{t},{\hat{I}}_{t}\right)=\left\{\begin{array}{c} 1,L_{ph}\left(I_{t},{\hat{I}}_{t}\right)<L_{ph}\left(I_{t},I_{s}\right)\\ 0,L_{ph}\left(I_{t},{\hat{I}}_{t}\right)\geq L_{ph}\left(I_{t},I_{s}\right) \end{array}\right..$$

The minimum projection technique proposed in [12] addresses the problem of occluded areas in the scene, which in essence, is also a mask, and we denote this mask by *M_m_* and calculate it by the following expression:(9)$$M_{m}\left(I_{t},{\hat{I}}_{t}\right)=\left\{\begin{array}{c} 1,L_{ph}\left(I_{t},{\hat{I}}_{t}\right)\leq \underset{s}{\min}L_{ph}\left(I_{t},I_{s}\right)\\ 0,L_{ph}\left(I_{t},{\hat{I}}_{t}\right)>\underset{s}{\min}L_{ph}\left(I_{t},I_{s}\right) \end{array}\right..$$

Then, all the masks are combined using element-wise logical conjunction, as shown below, to generate the final mask, marked *M_f_*.
(10)$$M_{f}\left(I_{t},{\hat{I}}_{t}\right)=M_{e}\left(I_{t},{\hat{I}}_{t}\right)\cdot M_{o}\left(I_{t},{\hat{I}}_{t}\right){\cdot M}_{a}\left(I_{t},{\hat{I}}_{t}\right){\cdot M}_{s}\left(I_{t},{\hat{I}}_{t}\right){\cdot M}_{m}\left(I_{t},{\hat{I}}_{t}\right),$$

Finally, the photometric consistency loss is updated as follows:(11)$$L_{ph}\left(I_{t},{\hat{I}}_{t}\right)=M_{f}\left(I_{t},{\hat{I}}_{t}\right)\cdot L_{ph}\left(I_{t},{\hat{I}}_{t}\right).$$

The final mask obtained through forward propagation also serves as a supervised signal for the MaskNet. The MaskNet takes the target-source image pairs as input and generates a mask of estimated outliers. Unlike the final mask, the values in the generated mask are not binary but continuous, ranging from 0 to 1, making it more convenient for model training. The MaskNet is trained by minimizing the difference between the estimated mask ($M_{e}$) and the calculated mask ($M_{f}$). The loss function is as follows:(12)$$L_{m}\left(M_{f},M_{e}\right)=-\frac{1}{n}\sum_{i}\left[M_{f_{i}}logM_{c_{i}}+\left(1-M_{c_{i}}\right)log\left(1-M_{f_{i}}\right)\right]$$
where *n* represents the number of elements in $M_{f}$ and $M_{e}$, and *log* is the logarithm function. Figure 5 shows two examples of the visualization of the calculated mask ($M_{f}$) and the estimated mask ($M_{e}$).

### 3.3. Geometric Consistency Loss 

To address the issue that the photometric consistency loss function fails to hold in scenes with significant illumination variations, we propose the geometric consistency loss with its generation mechanism illustrated in Figure 6. This loss leverages the geometric constraints inherent to the 3D scene, rendering it impervious to illumination variations and serving as a complementary measure to the photometric consistency loss. The geometric consistency loss comprises optical flow consistency loss, depth consistency loss, and pose consistency loss, which is expressed as follows:(13)$$L_{g}={\lambda_{f}L}_{flo}+{\lambda_{d}L}_{dep}+\lambda_{p}L_{pos}$$
with $L_{flo}$, $L_{dep}$ and $L_{pos}$ denoting the optical flow consistency loss, the depth consistency loss, and the pose consistency loss, respectively, and $\lambda_{f}$, $\lambda_{d}$ and $\lambda_{p}$ are the corresponding weighting coefficients.

To better extract the geometric features in the image when estimating the camera pose, we use the optical flow estimated by FlowNet as an input. In addition, the estimated optical flow ($F^{f}$) can provide an additional supervised signal to the framework. Specifically, using the estimated depth and the estimated camera pose, we can calculate the projected optical flow ($F^{cal}$) using the following expression: (14)$$F^{cal}\left(p_{t}\right)=KT_{t\to s}D_{t}\left(p_{t}\right)K^{-1}p_{t}-p_{t}.$$

After removing outliers, the calculated optical flow ($F^{cal}$) should be theoretically consistent with the estimated optical flow ($F^{f}$). Then the optical flow consistency loss can be calculated using the following formula:(15)$$L_{flo}=\sum_{p_{t}\in V}{\left\|F^{cal}\left(p_{t}\right)-F^{f}\left(p_{t}\right)\right\|}_{1}$$
where *V* denotes the valid region after excluding the outliers.

Since the correspondence between the target view and source view coordinates can be determined computationally during the projection process, there is also a correspondence between the target view depth map and the source view depth map estimated by DepthNet. That is, the source view depth map and the projected optical flow can inverse warp the target view depth map, which is consistent with the depth map estimated by DepthNet [18]. Therefore, we also use the L1 norm to calculate their differences, and the depth consistency loss is represented as follows:(16)$$L_{dep}=\sum_{p_{t}\in V}{\left\|D_{t}\left(p_{t}\right)-{\bar{D}}_{t}\left(p_{t}\right)\right\|}_{1}.$$

The pose consistency loss is obtained in the three-frame snippet by ensuring that the transformation matrices are closely coupled. Specifically, with the PoseNet network, the pose information between each pair of the three image frames can be estimated, denoted as $T_{t-1\to t}$, $T_{t\to t+1}$ and $T_{t-1\to t+1}$. Using the transformation relationship between them, i.e., $T_{t-1\to t}\cdot T_{t\to t+1}=T_{t-1\to t+1}$, the pose consistency loss is proposed to constrain the entire model further, expressed as follows: (17)$$L_{pos}=T_{t-1\to t}\cdot T_{t\to t+1}-T_{t-1\to t+1}.$$

## 4. Experiments

In this section, we conduct several experiments to evaluate the estimation results of our pose and depth models and visualize the estimated results. We also conduct ablation experiments to validate the effectiveness of our used strategies as well as tests on unfamiliar datasets to verify the generalization ability of the model.

### 4.1. Implementation Details

Our framework comprises four sub-networks. For the depth estimation network (DepthNet), we employ a U-shaped architecture [36] with skip connections, which takes a single image as input and outputs the corresponding depth map. The encoder mainly consists of cascaded residual convolutional neural networks, with ResNet18 [37] as the underlying network that contains 11 million trainable parameters. The decoder primarily consists of cascaded deconvolutional layers. For the pose estimation network (PoseNet), we also use ResNet18 as the encoder, followed by a global average layer before the final prediction to obtain the 6 DOF camera pose (3 for translation and 3 for rotation in Euler angles), with the pre-processed optical flow as input. The mask estimation network (MaskNet) has the same network structure as DepthNet but with different inputs, taking a stack of two frames in RGB channels and outputting the probability of each pixel being an outlier in the scene. Finally, for optical flow estimation, we adopt a pre-trained network, MaskFlownet [38], to accelerate the training process by computing the optical flow information for all adjacent and intermediate frames in advance, which can be directly read during training.

The framework was implemented on the PyTorch platform [39], and all experiments were performed using a single NVIDIA graphics card (RTX 1080 Ti) and an Intel Core i7 3.6 GHz CPU. We use Adam [40] for optimization. The hyperparameters of the loss function, including $\lambda_{s}=0.001$, $\lambda_{m}=0.2$, $\lambda_{f}=0.2$, $\lambda_{d}=0.2$ and $\lambda_{p}=0.5$. To ensure fair comparisons with other works, we cropped the image resolution to 640 × 192, accelerating the network training. Similar to other works [20,21], we applied data augmentation techniques such as cropping, random scaling, and horizontal flips. For all ResNet18-based models, including DepthNet, PoseNet, and MaskNet, we initialized their encoder parts using the weights pre-trained in ImageNet [41], following the practice of MonoDepth2 [12]. The batch size was set to 8, the initial learning rate was set to 0.0001, and it was multiplied by 0.6 every 5 epochs. The total training time was approximately 30 h, and the network was trained for 20 epochs.

### 4.2. Datasets and Metrics

We conducted our training and testing on the widely used KITTI benchmark [42], the most popular dataset in the field of autonomous driving. It provides 56 driving scenes at a rate of 10 frames per second with an image resolution of approximately 1226 × 370, covering various urban, residential, and highway driving scenarios. In addition, the dataset includes ground truth for camera poses and depth maps, which are derived from multiple modalities such as high-precision LiDAR, GPS, and IMU sensors. Since our method is unsupervised and only requires a sequence of consecutive frames as input, we also pre-trained our model on the Cityscapes dataset [43], which consists of video sequences of cars driving in over 50 cities and stereo data without annotations. Additionally, we validated the generalization performance of our trained model on the Make3D dataset [44], which includes single-view images and corresponding low-resolution depth maps but no monocular sequences or stereo images.

As in previous work [15], we adopted the absolute trajectory error (ATE) to evaluate pose estimation and the synthetic policy [12] to evaluate depth estimation. The following is the definition of ATE:(18)$$F_{i}=Q_{i}^{-1}SP_{i}\;ATE=\sqrt{\frac{1}{N}\sum_{i=1}^{N}{\left\|trans(F_{i})\right\|}^{2}}$$
*P_i_* and *Q_i_* represent the estimated pose and its corresponding ground truth, *S* denotes the similarity transformation matrix, and *trans* represents the translation component [45]. 

Depth estimation was evaluated using various metrics, including absolute relative error (*Abs. Rel*), which measures the relative error between predicted and ground truth values; square relative error (*Sq. Rel*), which squares *Abs. Rel* and accentuates the differences; root mean squared error (*RMSE*), which reflects the absolute error between predicted and ground truth values; log root mean squared error (*RMSE log*), which uses logarithmic operations to reduce the impact of outliers on *RMSE*; and prediction accuracy (*δ*), which intuitively reflects the accuracy of the predictions. The following are the definitions of these evaluation standards:(19)$$Abs.Rel=\frac{1}{N}\sum_{i=1}^{N}\frac{\left|D_{i}-D_{i}^{*}\right|}{D_{i}^{*}},$$
(20)$$Sq.Rel=\frac{1}{N}\sum_{i=1}^{N}\frac{{\left|D_{i}-D_{i}^{*}\right|}^{2}}{D_{i}^{*}},$$
(21)$$RMSE=\sqrt{\frac{1}{N}\sum_{i=1}^{N}{\left|D_{i}-D_{i}^{*}\right|}^{2}},$$
(22)$$RMSE(log)=\sqrt{\frac{1}{N}\sum_{i=1}^{N}{\left|{lgD}_{i}-{lgD}_{i}^{*}\right|}^{2}},$$
(23)$$\delta =max(\frac{D_{i}}{D_{i}^{*}},\frac{D_{i}^{*}}{D_{i}})<T,$$
where $D_{i}$ and $D_{i}^{*}$ represent the estimated depth and its related ground truth. We used *T* values of 1.25, 1.25^2^, and 1.25^3^. Lower values for error metrics (*Abs. Rel*, *Sq.Rel*, *RMSE*, *RMSE log*) and higher values for the accuracy metric (*δ*) indicate better performance.

### 4.3. Evaluation of Depth Estimation

In the monocular depth estimation experiments, we evaluated the performance of our depth estimation network using the widely used Eigen split of the KITTI dataset [42]. Specifically, our training set consisted of 39,810 monocular triplets, while the validation set consisted of 4424 triplets, and the testing set included 697 representative frames. As with other unsupervised methods, the median ratio [15] aligns each predicted depth map with the corresponding ground truth depth map. Again, conventional metrics and the cropping region in [12] were used, and the upper limit of the standard depth was set to 80 m. Finally, we compared our method with other classic approaches using learning-based methods, and the quantitative and qualitative comparisons are shown in Table 1 and Figure 7.

In the supervised signal column of Table 1, “Depth” indicates that the method is supervised, “Stereo” suggests that the method is trained on stereo images with baseline information, and “Mono” suggests that the method is trained solely on monocular image sequences. The third column indicates the dataset used for training, where “K” denotes training only on the KITTI dataset and “CS + K” denotes fine-tuning on the KITTI dataset following pre-training on the Cityscapes dataset. Among all the unsupervised methods, our method outperforms all the others, especially Godard’s Monodepth2 [12], a classical depth estimation network. Our model has the same number of parameters compared to Monodepth2’s, but the model’s performance is significantly improved by using a variety of optimization strategies we have proposed. We also pre-trained our model on the Cityscapes dataset [43] and fine-tuned it on the KITTI dataset. The results (in the bottom part of Table 1) show that increasing the training data can improve the model’s performance. This demonstrates the advantage of unsupervised methods over supervised methods, where the estimation accuracy increases as the training data grows and the model size increases. However, for supervised methods, increasing the training data requires more work to generate labels, and increasing the model capacity blindly without increasing the data can lead to overfitting. This is why unsupervised methods are increasingly preferred by researchers. 

A qualitative comparison of our method with some classical methods is shown in Figure 7. Compared with other methods, our method estimates the boundaries of various objects more clearly, including dynamic objects (moving cars and pedestrians) and distant cars.

### 4.4. Evaluation of VO Estimation

For pose estimation, we conducted experiments on the KITTI odometry dataset, using sequences 00-08 for training and sequences 09 and 10 for testing, following the method of previous work [15]. To ensure a fair comparison, the input sequences were modified from 3 to 5 frames to be consistent with other methods [11,12,13]. Our method’s qualitative and quantitative results compared to similar methods are shown in Figure 8 and Table 2, respectively. 

Our method achieves the best results, especially over Zhou et al. [15]. On the one hand, they use raw RGB images in estimating the pose, which contains redundant information that does not help the network learn the motion information of the camera. On the other hand, many outliers in the scene, such as occlusions and dynamic objects, impact their pose estimation model. In contrast, our method uses optical flow as input, which contains camera motion information that can be more easily learned by the network. In addition, the interference of outliers, such as dynamic objects, is effectively avoided by the multiple masking techniques, which leads to a significant improvement in the accuracy of pose estimation.

### 4.5. Ablation Study

To verify the effectiveness of our proposed strategies, we conducted an ablation study on the entire framework. Since the DepthNet and the PoseNet adopt a joint training method and their accuracy also affects each other, it is sufficient to test only one sub-network. Here we test the PoseNet, using the sequences 00-08 on the KITTI dataset as the training set and the sequences 09 and 10 as the test set. The results of the ablation experiments are shown in Table 3. 

The baseline model represents that no mask is used, and the loss function used for training only includes the photometric consistency loss and the smoothness loss. $M_{f}$ denotes that the model calculated the mask used to eliminate outliers during training and used the mask when calculating the photometric consistency loss. The model is represented by $M_{e}$ not only calculates the mask $M_{f}$ during training but also adds *L_m_* to the final loss function to train the MaskNet and preprocesses the input of PoseNet with the estimated mask $M_{e}$. $L_{flo}$ represents that only the optical flow consistency constraint is used, $L_{dep}$ represents that only the depth consistency constraint is used, $L_{pos}$ represents that only the pose consistency constraint is used, and $L_{full}$ represents a complete model. That is, two masks and three consistency constraints are used. 

We evaluate each improved component in the proposed monocular system and remove them from the whole system to prove their effectiveness indirectly. It can be seen from Table 3 that the accuracy of the baseline is the worst of all models, and all the strategies we propose help to improve the accuracy of the framework. In addition, we found that the increase of $L_{dep}$ and $L_{pos}$ was relatively small, while the increase of $L_{flo}$ was relatively large. We believe this is because the optical flow consistency constraint, the supervised signal used, comes from the trained network. This additional supervised signal has further improved the accuracy of the model. Depth consistency and pose consistency are implicit constraints within the model. Their design is mainly used to ensure global consistency in pose estimation. Since the input is a continuous multi-frame picture, the scale consistency constraint of every two frames will also ensure that the constant multi-frame pictures maintain the same scale, thus ensuring the global consistency of pose estimation.

### 4.6. Generalization on Make3D Dataset

To verify the generalization of our trained model, we conducted tests on the Make3D [44] dataset using the model trained on the KITTI and Cityscapes datasets. This means that our model has never encountered any images from the Make3D dataset before. The qualitative and quantitative results of the depth estimation network are shown in Figure 9 and Table 4. It can be observed that our method has certain advantages over other methods of the same type. However, due to different domain biases in different datasets, there is still room for improvement in the performance of our model compared to the results obtained on the KITTI dataset.

## 5. Conclusions

This paper introduces a novel unsupervised learning framework for estimating scene depth and camera pose from video sequences, focusing on challenging scenes. Our proposed method incorporates multiple mask techniques to identify and eliminate the influence of outliers in challenging scenes during the view synthesis process and pose estimation. Furthermore, we have proposed several geometrically consistent loss functions as additional supervised signals to enhance the performance of our model. We conducted evaluation and ablation experiments on the KITTI dataset, and the results validate the effectiveness of our contributions. Our framework has promising potential for addressing the challenges of estimating scene depth and camera pose in real-world scenarios. Future work can extend our method to handle more complex scenes and design the framework using high-capacity models such as Transformer and training with more data.

## Figures and Tables

**Figure 1 sensors-23-05329-f001:**
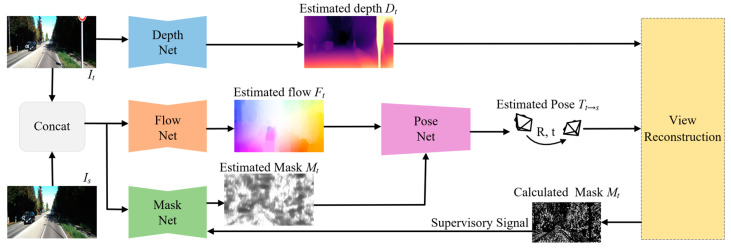
Overview of our method. The input contains two adjacent images. The DepthNet estimates the depth map of the target view (*I_t_*). The FlowNet estimates the optical flow with the target view (*I_t_*) and the source view (*I_s_*). The MaskNet also uses the target-source image pairs as input to estimate the outliers in the scene. The estimated mask is used to preprocess the optical flow, and then the preprocessed optical flow is used as the input of the PoseNet. The estimated depth (*D_t_*), camera pose ($T_{t\to s}$), and the source view (*I_s_*) are used to synthesize a new target view ($\hat{I}$*_t_*). Then the DepthNet and the PoseNet are trained by minimizing the difference between the synthesized view ($\hat{I}$*_t_*) and the target view (*I_t_*) as the main supervised signal. Moreover, the mask obtained by calculation in the process of forward propagation is also used as the supervised signal for the MaskNet.

**Figure 2 sensors-23-05329-f002:**
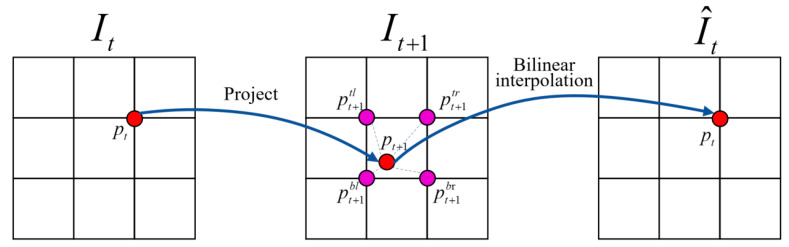
Illustration of the view syntheses process. The projection process involves using the estimated depth and camera pose to calculate the position of each point *p_t_* in the target view *I_t_* projected onto the source view *I_t_*_+1_. The bilinear interpolation means that the pixel value of each point on the synthesized view ${\hat{I}}_{t}$ is calculated by using bilinear interpolation on the source view *I_t_*_+1_.

**Figure 3 sensors-23-05329-f003:**
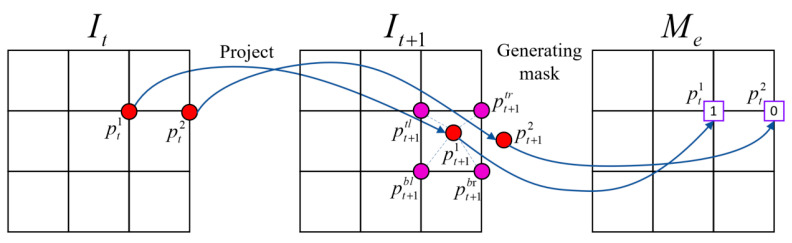
The process of generating the boundary mask.

**Figure 4 sensors-23-05329-f004:**
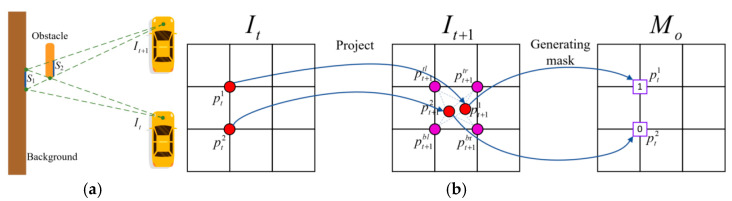
(**a**) Effect of occluded areas on view synthesis process. (**b**) The process of generating the occlusion mask.

**Figure 5 sensors-23-05329-f005:**
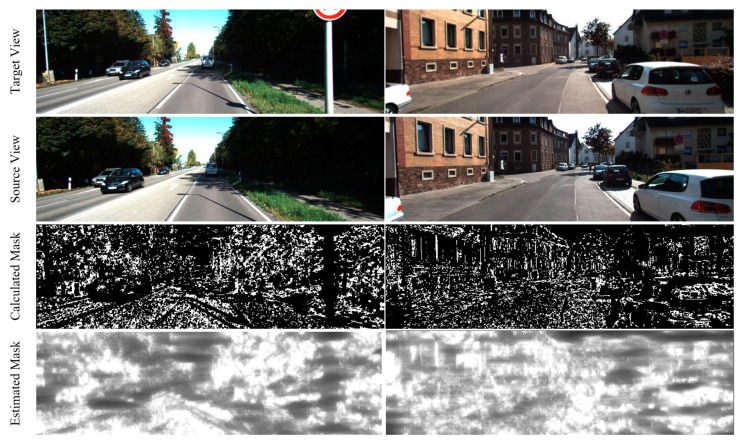
Two examples of our proposed mask visualization. The goal of the calculated and estimated masks is to mitigate the negative impact of challenging scenes on the view synthesis process and VO estimation, respectively.

**Figure 6 sensors-23-05329-f006:**
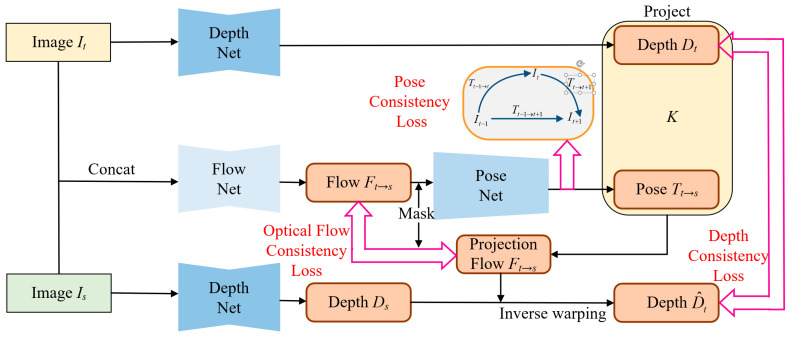
The generation mechanism of the geometric consistency loss function. The geometric consistency loss consists of three components: optical flow consistency loss, depth consistency loss, and pose consistency loss. The optical flow consistency loss is calculated from the difference between the estimated optical flow and the calculated projected optical flow; the depth consistency loss is calculated from the difference between the estimated depth and the inverse warping depth [18]; and the pose consistency loss is obtained in three frames of snippets by ensuring a tight coupling of the transformation matrices with each other.

**Figure 7 sensors-23-05329-f007:**
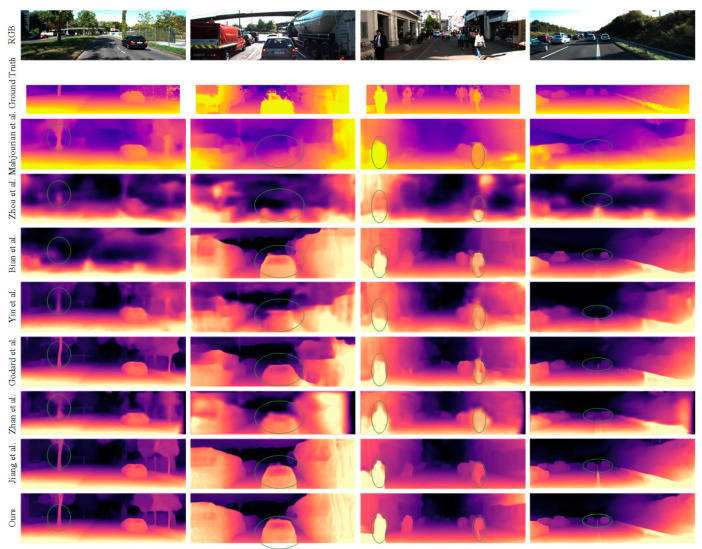
Our approach is compared qualitatively with that of Mahjourian et al. [27], Zhou et al. [15], Bian et al. [18], Yin et al. [31], Godard et al. [12], Zhan et al. [13], and Jiang et al. [21].

**Figure 8 sensors-23-05329-f008:**
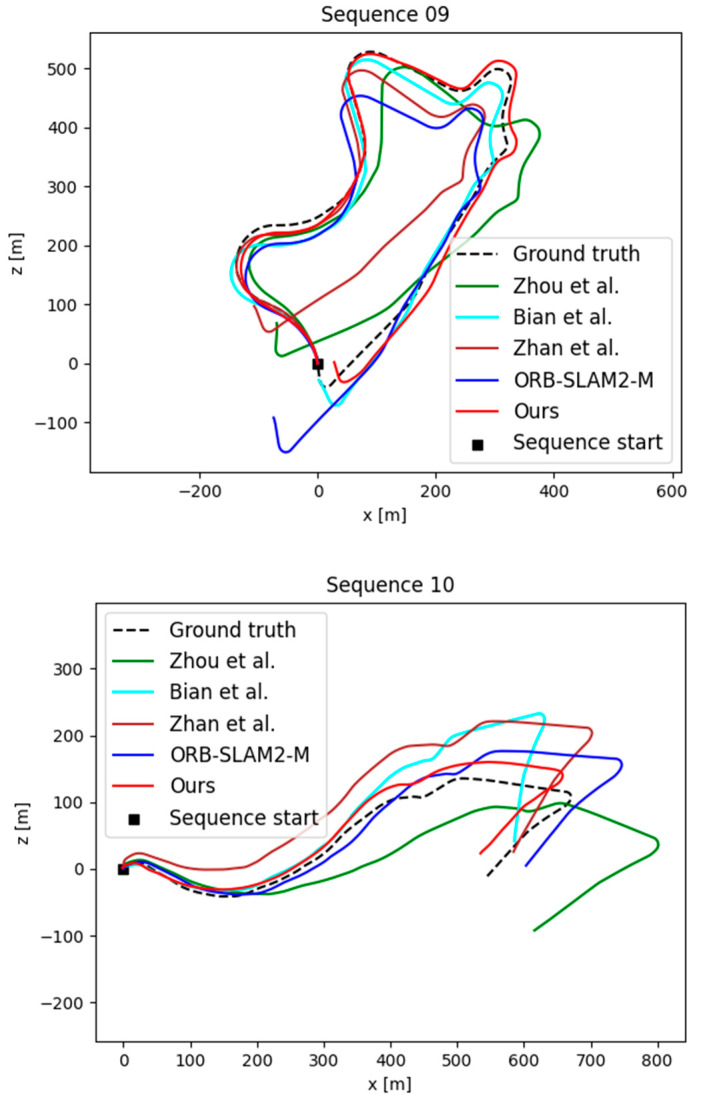
Comparison of trajectories generated by our method and the methods of Zhou et al. [15], Bian et al. [18], Zhan et al. [13], and ORB-SLAM2-M [6] on the KITTI sequences 09 and 10.

**Figure 9 sensors-23-05329-f009:**
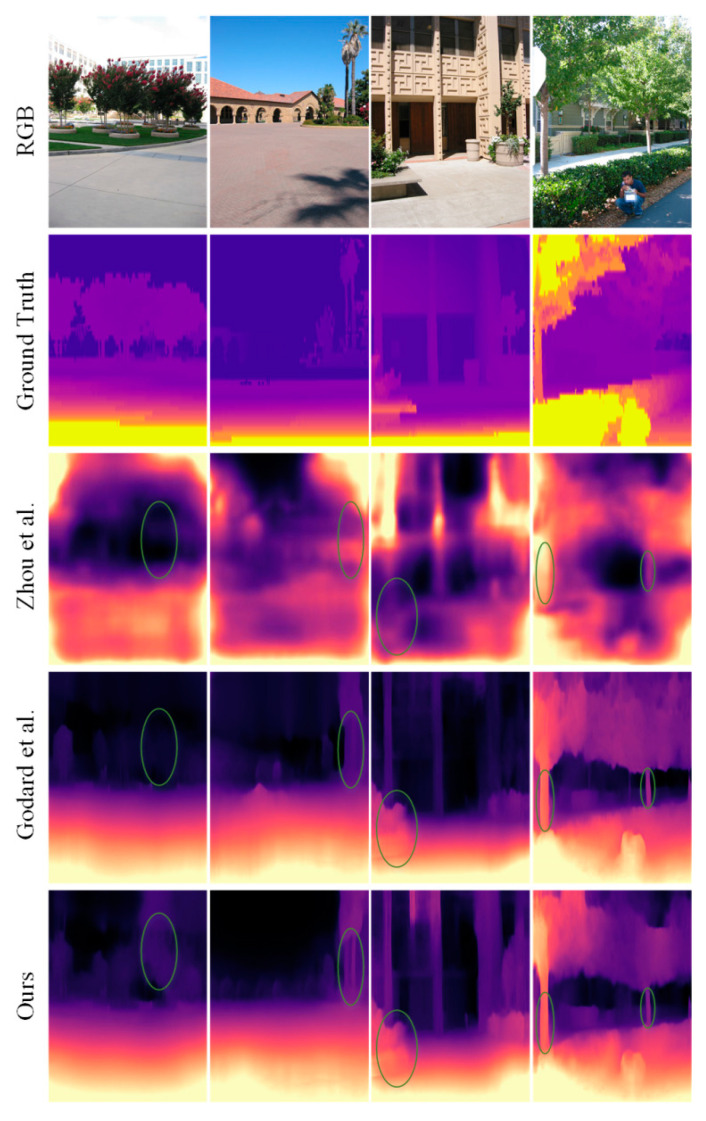
A qualitative comparison of our method with that of Zhou et al. [15] and Godard et al. [12] on the Make3D dataset.

**Table 1 sensors-23-05329-t001:** The quantitative comparison of our method with other methods.

Method	Supervised Signal	Training Dataset	Error Metric				Accuracy Metric		
			Abs.Rel	Sq.Rel	RMSE	RMSE (log)	δ<1.25	$\delta <1.25^{2}$	$\delta <1.25^{3}$
Eigen et al. [22] Coarse	Depth	K	0.214	1.605	6.563	0.292	0.673	0.884	0.957
Eigen et al. [22] Fine	Depth	K	0.203	1.548	6.307	0.282	0.702	0.890	0.958
Liu et al. [23]	Depth	K	0.202	1.614	6.523	0.275	0.678	0.895	0.965
Zhan et al. [13]	Stereo	K	0.144	1.391	5.869	0.241	0.803	0.928	0.969
Godard et al. [12]	Stereo	K	0.148	1.344	5.927	0.247	0.803	0.922	0.964
Zhou et al. [15]	Mono	K	0.208	1.768	6.856	0.283	0.678	0.885	0.957
Zhou et al. [15] updated	Mono	K	0.183	1.595	6.709	0.270	0.734	0.902	0.959
Mahjourian et al. [27]	Mono	K	0.163	1.240	6.220	0.250	0.762	0.916	0.968
Jin et al. [28]	Mono	K	0.169	1.387	6.670	0.267	0.748	0.904	0.960
Yin et al. [31]	Mono	K	0.155	1.296	5.857	0.233	0.793	0.931	0.973
Ranjan et al. [34]	Mono	K	0.140	1.070	5.326	0.217	0.826	0.941	0.975
Wang et al. [20]	Mono	K	0.147	0.889	4.290	0.214	0.808	0.942	0.979
Bian et al. [18]	Mono	K	0.137	1.089	5.439	0.217	0.830	0.942	0.975
Jiang et al. [21]	Mono	K	0.112	0.875	4.795	0.190	0.880	0.960	0.981
Godard et al. [12]	Mono	K	0.154	1.218	5.699	0.231	0.798	0.932	0.973
Ours	Mono	K	0.110	0.793	4.553	0.184	0.886	0.964	0.983
Zhou et al. [15]	Mono	CS + K	0.198	1.836	6.565	0.275	0.718	0.901	0.960
Mahjourian et al. [27]	Mono	CS + K	0.159	1.231	5.912	0.243	0.784	0.923	0.970
Jin et al. [28]	Mono	CS + K	0.162	1.039	4.851	0.244	0.767	0.920	0.969
Yin et al. [31]	Mono	CS + K	0.153	1.328	5.737	0.232	0.802	0.934	0.972
Ranjan et al. [34]	Mono	CS + K	0.139	1.032	5.199	0.213	0.827	0.943	0.977
Wang et al. [20]	Mono	CS + K	0.155	1.184	5.765	0.229	0.790	0.933	0.975
Ours	Mono	CS + K	0.103	0.634	3.367	0.178	0.899	0.969	0.984

**Table 2 sensors-23-05329-t002:** The performance of various methods on the KITTI VO dataset was compared in terms of Absolute Trajectory Error (ATE).

Method	ATE of Seq.09	ATE of Seq.10
ORB-SLAM [6]	0.014 ± 0.008	0.012 ± 0.011
Zhou et al. [15]	0.016 ± 0.009	0.013 ± 0.009
Bian et al. [18]	0.016 ± 0.007	0.015 ± 0.015
Mahjourian et al. [27]	0.013 ± 0.010	0.012 ± 0.011
Yin et al. [31]	0.012 ± 0.007	0.012 ± 0.009
Ranjan et al. [34]	0.012 ± 0.007	0.012 ± 0.008
Ours	0.008 ± 0.005	0.007 ± 0.005

**Table 3 sensors-23-05329-t003:** Comparison between various variants of our model on KITTI odometry.

Method	ATE of Seq.09	ATE of Seq.10
Baseline	0.018 ± 0.011	0.017 ± 0.011
$\mathrm{Baseline}+M_{f}$	0.014 ± 0.010	0.013 ± 0.010
$\mathrm{Baseline}+M_{e}$	0.012 ± 0.008	0.011 ± 0.008
$\mathrm{Baseline}+M_{e}$ + $L_{dep}$	0.012 ± 0.007	0.010 ± 0.007
$\mathrm{Baseline}+M_{e}$ + $L_{pos}$	0.011 ± 0.009	0.009 ± 0.006
$\mathrm{Baseline}+M_{e}$ + $L_{flo}$	0.009 ± 0.005	0.007± 0.005
$\mathrm{Baseline}+M_{e}$ + $L_{full}$	0.008 ± 0.005	0.007 ± 0.005

**Table 4 sensors-23-05329-t004:** Quantitative comparison of depth estimation of different methods on Make3D dataset.

Method	Error Metric			
	Abs. Rel	Sq. Rel	RMSE	RMSE (log)
Liu et al. [23]	0.481	6.761	10.55	0.169
Zhou et al. [15]	0.396	5.731	10.869	0.513
Godard et al. [12]	0.579	11.235	11.892	0.201
Ours	0.304	3.452	7.186	0.203

## Data Availability

The data are available in a publicly accessible repository.

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
