# Peer review of "Unsupervised Monocular Depth and Camera Pose Estimation with Multiple Masks and Geometric Consistency Constraints"

_sensors, 2023, doi:10.3390/s23115329_

Round 1
Reviewer 1 Report
I would like to highlight some aspects of the paper that need to be corrected to improve the understanding of the technique:
In "abstract":
You say: "In addition, since challenging scenes can also impact pose estimation, the calculated mask is utilized as a supervised signal to train a mask estimation network for preprocessing the input to the pose estimation network, which aimed to improving the estimation accuracy". I have two points:
(1) The first point is that the sentence is long.
(2) And second point is in "...estimation accuracy". What is "...estimation accuracy" about ?
In Section "1. Introduction"
In the following sentence: "Traditional methods rely on geometric cues in the image for inference, making them sensitive to challenging environments with low texture or strong lighting changes".
(3) Please add citations.
In the following sentence: "Conversely, learning-based methods for depth and pose estimation exhibit better adaptability to challenging environments."
(4) Please add citations from books or journals.
In the folowing sentence: "their reliance on massive quantities of labeled data, which can be highly depends on the accuracy of manual annotation [4], prevents them from further improving the performance of their models" sounds strange for me.
(5) I suggest a change: "...their reliance on massive quantities of labeled data depends on the accuracy of manual annotation that prevents them from further improving the performance of their models."
(6) In the following sentence "...quantities of labeled data, which can be highly depends on the accuracy". I think the construction 'be depends' is wrong".
(7) In the following sentence: "As a result, any degradation in pose estimation accuracy will result in a decrease in total performance". What do you mean by total performance? What performance are you referring to?
(8) I didn't notice any numerical results that summarize all your results in the abstract or in Introduction.
(9) You say "We evaluate our model on the widely used KITTI dataset and demonstrate its superiority over other unsupervised methods." How much is this superiority?
(10) Put the structure of the document in the last paragraph of this section. Section x describes a... Section y details b...
In Section "2. Related Works"
(11) In "Garg et al. [7] pioneered a novel method to reduce the reliance on labeled data by". Does the word reliance mean dependence ? I suggest to change to avoid ambiguity.
(12) In "solved the absolute scale problem of VO estimation" Add a citation that explains the absolute scale problem.
(13) You say "3D geometry of the whole scene and required that the estimated 3-dimensional". Don't 3D and 3-dimensional mean the same thing? It's good to standardize.
In Section "3. Methods"
(14) Put an introductory paragraph between the "Methods" section and the "Method Overview" section
In Section "3.1. Method Overview"
(15) You say: "while exhibiting good robustness to outliers in the scene, such as dynamic objects and occlusion areas". You did not still define dynamic objects or occlusion areas. If you describe further ahead, I suggest to point to this.
(16) The first paragraph of this Section (3.1. Method Overview) is too long. Break into two or more.
(17) In "...PoseNet's input; and a geometric consistency loss, which provides an additional weak supervisied signal" Fix "supervisied" to supervised.
(18) Put a comma after Equation 1.
(19) In "l is the scale...". Do not start the sentence with an Equation letter. One option could be: "The parameter l is the scale..."
(20) In "...used as the supervisied signal for the MaskNet". Fix "supervisied" to supervised.
(21) In "The estimated depth (Dt), camera pose (Tts), and the source view (Is) are used to synthesize a new target view (^Is)". I can't verify this sentence in the image. Could you detail the image more to contain this sentence?
In Section "3.2. Photometric Consistency Loss and Smoothness Loss"
(22) In the following sentence: "which is expressed as". Maybe at the end you can put a colon "which is expressed as:" Check this in the other sentences.
(23) Put a comma after Equation 2.
(24) In Equation 3 you use a dot to denote multiplication, but you also join two variables, without a dot, to denote multiplication. Standardize the way you define multiplication in all Equations.
(25) Put a comma after Equation 4.
(26) In "...of the reconstructed frame t based on the projected position on frame t+1." Do the terms t and t+1 denote image? Wouldn't it be I_t and I_{t+1} ?
(27) You say: "The photometric value of the synthesized view is theoretically consistent with that of the target view". Why theoretically consistent?
(28) Put a comma after Equation 5.
(29) Is SSIM a known equation? I suggest you define the SSIM Equation in the article, for completeness.
(30) Could you explain Equation 6 better? Is the term D the same as in Equation 2?
(31) Suggestion: In Figure 2 it is good to describe the resolution of the image. For example, "This image has a resolution of 3 x 3 pixels. I suggest also entering the coordinates of the image.
In Section "3.3. Calculated mask and mask loss"
(32) You say "...some pixels will be projected beyond the imaging plane of the source view, so that the pixel value of the point cannot be reconstructed" Why are pixels projected outside the image plane?
(33) In "...When the camera moves to the frame t+1" Does the term t+1 denote image? Wouldn't it be I_{t+1} ?
(34) In "while for some pixel points caused by various adverse factors, the photometric error value will always remain at a higher value". Are there any examples that demonstrate this effect?
(35) Put a comma after Equation 7.
(37) In the sentence: "...where Lph is the average value of all photometric error" Could you explain better? Is this averaging performed over the entire image?
(38) In the sentence "1.5 is used here empirically". Is there any relationship between weight and training capacity.
(39) In the sentence: "...of which violate the assumptions required for the view synthesis process". Reiterate what the assumptions are.
(40) In the sentence: "Unlike the other calculated masks, the values in this mask are not binary but continuous, ranging from 0 to 1". Which mask are you referring to, the final mask or the mask generated by MaskNet?
In Section "3.4. Geometric consistency loss"
(41) In Figure 6 you say: "the depth consistency loss is calculated from the difference between the estimated depth and the inverse warping depth". Could you explain what "inverse warping depth" is, or cite a book that explains it?
(42) In the sentence: "...should be theoretically consistent with the estimated optical flow..." . What do you mean by consistent? detail more.
In Section "4 Experiments"
(43) Put introductory paragraph between the "Experiments" section and the "Implementation Details" section
In Section "4.1. Implementation Details"
(42) In the sentence "The total training time was approximately 30 hours, and the network was trained for 20 epochs." I advise putting a graph demonstrating the convergence of the training phase.
(43) In the sentence "...at a frame rate of 10" change to "...at a rate of 10 frames per second".
In Section "4.2. Datasets and Metrics"
(44) In the sentence: "where P i and Q i represent the estimated pose and its corresponding ground truth, S denotes the similarity transformation matrix, and trans represents the translation component." Detail more what is similarity transformation matrix. Put a quote about the concept.
In Section "4.3. Evaluation of Depth Estimation"
(45) In the sentence: "We used T 1.25..." Why did you use these values?
(46) In Table 1, put the best result in each column in bold.
(47) Break that first paragraph into three smaller paragraphs.
(48) Do not finish the section with an image or table. Finish always with text.
In Section "4.4. Evaluation of VO Estimation"
(49) In the sentence: "Our method achieves the best results among all methods, especially over Zhou et al. [8]." Why especially Zhou et al?
(50) Do not finish the section with an image or table. Finish with text.
In Section "4.5. Ablation Study"
(51) Quebre esse primeiro paragrafo em três paragrafos menores.
(52) Do not finish the section with an image or table. Finish with text.
(53) You do not mention the word ablation throughout the text. Introduce that term before.
General comments:
(54) I suggest you put a glossary to facilitate the search for symbols.
(55) what is multi-view synthesis (you mention it in related works) ?
(56) what is rigid regions (you mention it in related works) ?
(57) Equations are difficult to understand.
English is fine.
Reviewer 2 Report
The method used by the authors of four sub-networks and the results obtained show, comparatively, that there is a significant improvement in the determination of the depth of the images and the camera pose.
The step-by-step description and the well-presented detail of the elements involved in the training allow us to understand how and why they are incorporated into the formulas used for the calculation of the losses.
Comparisons with the results of some influential authors in the field show that they have considered the several unsupervised methods that have been proposed for monocular depth estimation, and image processing.
Author Response
Thank you for your positive feedback on our method and results. We value your feedback and are grateful for your recognition of our contributions. If you have any further comments or suggestions, please feel free to share them with us.
Thank you once again for your time and expertise in reviewing our work.
Reviewer 3 Report
Title of the paper: "Unsupervised Monocular Depth and Camera Pose Estimation with Multiple Masks and Geometric Consistency Constraints"
As a researcher working in the same field, I am impressed by the technique introduced in the paper, because it sheds new light on the earlier results of several authors and obviously can be successfully used in practice. From this point of view, the subject of the paper fits well with the scope of the journal (Sensors).
The paper is ended with numerical simulations that corroborate the theoretical results.
This manuscript contains new ideas and good results that help other researchers.
The decision is too a major revision for publication in the "Sensors".
Therefore, I recommend publishing this work after taking these points into account.
1- The introduction needs to explain the main contributions of the work more clearly.
2- The novelty of this paper is not clear. The difference between the present work and previous Works should be highlighted.
3- In the references in the Introduction section, the authors cite some works. However, they have not indicated the advantage or disadvantage and their relations to this paper. It’s a little confusing.
4- Need a detailed explanation of the preprocessing steps.
5- It is necessary to pay attention to the fact that before numerically solving this problem, it would be necessary to formulate a clear statement about its solvability and, since the authors are talking about multiple solvabilities, it is necessary to indicate how many solutions exist.
6- More motivation/context regarding the application side of it, particularly on the aspects that make this technique particularly suited for industrial/medical application scenarios, and how it would be applied in real scenarios. These aspects could additionally be supported with some related work context.
7- The authors can add the following reference to enrich the introductory section:
*A numerical method for solving the nonlinear equations of Emden-Fowler models, Journal of Ocean Engineering and Science, 2022. doi.org/10.1016/j.joes.2022.04.019.
Sincerely Yours
Minor editing of English language required.
Round 2
Reviewer 3 Report
No comments
Minor editing of English language required